



# Constraining a land cover map with satellite-based aboveground biomass estimates over Africa

Guillaume Marie[1], B. Sebastiaan Luysseart[2], Cecile Dardel[3], Thuy Le Toan[4], Alexandre Bouvet[4], Stéphane Mermoz[4], Ludovic Villard[4], Vladislav Bastrikov[5], Philippe Peylin[1].

5  [1]Laboratoire des Sciences du Climat et de l'Environnement (LSCE/IPSL), CEA-CNRS-UVSQ, Université Paris-Saclay, Gif-sur-Yvette, France
[2]Faculty of Science, Vrije Universiteit Amsterdam, Amsterdam, The Netherlands
[3]Laboratoire Géosciences Environnement, Paul Sabatier University, Toulouse III, Toulouse, France
[4]Centre d'Etudes Spatiales de la Biosphère (CESBIO), Toulouse, France
10  [5]Science Partners, Paris, France

*Correspondence to*: Guillaume Marie (Guillaume.Marie@uantwerpen.be)

**Abstract.** Most land surface models can either calculate the vegetation distribution and dynamics internally by making use of biogeographical principles or use vegetation maps to prescribe spatial and temporal changes in vegetation distribution. Irrespective of whether vegetation dynamics are simulated or prescribed, it is not practical to represent vegetation across the globe at the species level because of its daunting diversity. This issue can be circumvented by making use of 5 to 20 plant functional types (PFT) by assuming that all species within a single functional type show identical land–atmosphere interactions irrespective of their geographical location. In this study, we hypothesize that remote-sensing based assessments of above-ground biomass can be used to refine discretizing real-world vegetation in PFT maps. Remotely sensed biomass estimates for Africa were used in a Bayesian framework to estimate the probability density distributions of woody, herbaceous, and bare soil fractions for the 15 land cover classes, according to the UN-LCCS typology, present in Africa. Subsequently, the 2,5 and 97,5 percentile of the probability density distributions were used to create 2,5% and 97,5% confidence interval PFT maps. Finally the original and refined PFT maps were used to drive biomass and albedo simulations with the ORCHIDEE model. This study demonstrates that remotely sensed biomass data can be used to better constrain PFT maps. Among the advantages of using remotely sensed biomass data were the reduced dependency on expert knowledge and the ability to report the confident interval of the PFT maps. Applying this approach at the global scale, would increase confidence in the PFT maps underlying assessments of present day biomass stocks.

## 1 Introduction

Degradation, fires and deforestation of tropical forests are responsible for two thirds of the global net deforestation emissions [Houghton et al. 2012; Le Quéré et al. 2015; Friedlingstein et al. 2020]. Although African tropical rainforests represent only 30  one third of the global tropical rainforests [Lewis et al. 2009], they were responsible for almost all, i.e. 1,48 PgC in 2015 and





1,65 PgC in 2016, of the net C emissions of pan-tropical regions, but substantial uncertainty is associated with these estimates , i.e., 1,15 for 2015 and 1,0 PgC for 2016, mainly driven by fire and land use changes [Palmer et al. 2019]. The uncertainty of model estimates, such as mentioned above, broadly comes from three sources: (1) the vegetation distribution in the model, (2) the ability of the model to simulate biomass accumulation of undisturbed vegetation, and (3) the ability of

the model to simulate natural and anthropogenic disturbances of the standing biomass. As this study will focus on improving the description of the vegetation distribution, the first question that needs to be answered is why vegetation distribution remains so uncertain?

Most land surface models can either calculate the vegetation distribution internally by making use of biogeographical principles [Sitch et al. 2003, Krinner et al. 2005, Clark et al. 2011] or use vegetation maps to prescribe spatial and temporal

changes in vegetation distribution. Where the first approach results in a description of the potential vegetation, the second approach is more suitable when actual vegetation is to be studied. Irrespective of whether potential or actual vegetation is studied, it is not practical to represent vegetation across the globe at the species level because there are already over 60,000 tree species [Beech et al. 2017], not to mention the diversity in herbs, forbs and mosses. Land surface models represent this daunting diversity by making use of 5 to 20 plant functional types (PFT) [Huete et al. 2016]. The underlying assumption of

plant functional types is that all species within a single functional type show identical land–atmosphere interactions irrespective of their geographical location [Huete et al. 2016, Bonan et al. 2002, Brovkin et al. 1997, Chapin et al. 1996]. Discretizing real-world vegetation in PFTs is a first source of uncertainty.

When actual vegetation is the focus of a modelling study, the vegetation distribution will have to be prescribed. The construction of vegetation maps first requires real-world observations, typically through satellite-based remote sensing.

Current remote sensing technology does not enable distinguishing individual tree species, hence, vegetation is observed as land cover types [Defourny, P., 2019] which group vegetation with similar sensory characteristics. Remote sensing observations as well as classifying them in land cover types is a second source of uncertainties [Hansen et al. 2013, Mitchard et al. 2014, Hurtt et al. 2004]. Because the land surface models require the vegetation to be discretized in PFTs, which may differ between different land surface models, the land cover types will have to be remapped on PFT maps. The rules applied

in remapping satellite-based land cover types in PFT maps is formalized in so-called "cross-walking tables" (CWT) [Poulter et al. 2011, Poulter et al. 2015] which are a third source of uncertainty [Hartley et al., 2017].

Although CWTs have been extensively used to create PFT maps [Wei et al. 2018, Wei et al. 2016, Poulter et al. 2011, Krinner et al 2005], the process of associating land cover types with specific PFTs remains difficult to reproduce since several iterations of expert knowledge are required [Poulter et al. 2011, Poulter et al. 2015]. Various land cover

classifications exist, in particular the commonly used FAO (Food and Agriculture Organization) Land Cover Classification System (LCCS, Di Gregorio and Jansen, 2000). Most classes of the LCCS correspond to a mix of PFTs, which fractions are difficult to assess and likely variable across regions. For example several classes are labeled as a mosaic of vegetation types (i.e. "Mosaic of natural vegetation (tree, shrubs, herbs)"; see Table 2 in Poulter et al. 2015). Not surprisingly, efforts have been made to decrease the need of expert knowledge in favor of more objective and reproducible approaches, e.g.,





classification rules based on a suite of improved and standard MODIS products [Wanxiao et al. 2008]. Moreover, producing PFT maps from satellite-based land cover maps needs to become fully automated when the temporal frequency of satellite-based land cover and biomass maps increases, i.e., the GEDI [Dubayah et al. 2020] and BIOMASS missions [Le Toan et al. 2011, Quegan et al. 2019].

In this study, we hypothesize that remote-sensing based assessments of above ground biomass (ABG) can decrease the
dependency on expert knowledge when setting up CWTs and as such contribute to the automation of the land cover class mapping into PFTs for land surface models. The main rationale is that the above-ground biomass content of an ecosystem provides information on the fraction of tree PFTs of that ecosystem. In this context, the objective of this study are: (1) construct a framework of data assimilation in which biomass remote sensing products can be routinely used to update an existing or create a new CWT, (2) refine a cross-walking table used to convert the ESA-CCI Global Land Cover map into a
PFT map, and (3) propagate the confident interval from using a CWT in the production of PFT maps, to the simulation results of a land surface model. Such a framework will be applied and tested over Africa using the above ground biomass product derived by [Bouvet et al 2018] for that continent with the ORCHIDEE land surface model [Krinner et al., 2005] and the version used for the recent Climate Modelling Intercomparison Project - phase 6 (CMIP6) [Boucher et al., 2020].

## 2 Materials and methods

**2.1 Overview**

Cross-walking tables (CWT) [Poulter et al. 2015] are used to convert the 43 land cover types distinguished on the ESA-CCI land cover product into generic plant functional types (13 PFTs in Poulter et al., 2015) distinguished by large-scale land surface models such as the ORCHIDEE model [Krinner et al., 2005] used in this study. These generic PFTs are further grouped and/or divided to match each model-specific PFT classification, using additional grid-cell information to separate
grassland and crop C3 versus C4 photosynthetic pathway [Still et al., 2003] and to split generic PFT according to bioclimatic zones (i.e., Koppen Geiger climate classification map) (see more details for the ORCHIDEE model in Lurton et al, 2020). In this study, we will create a new ORCHIDEE PFT map by combining information from the ESA-CCI land cover product and the AGB product for Africa [Bouvet et al 2018] to estimate woody, herbaceous and bare soil cover fractions within each land cover type of the ESA-CCI product. Subsequently, the estimated cover fractions are used to refine the existing CWT and
create a new ORCHIDEE PFT map applicable primarily for Africa (Fig. 1). Finally, the impact of using AGB maps to refine the PFT maps on the skill of the ORCHIDEE model to simulate the contemporary biomass and its spatial distribution over Africa is quantified. Note that the approach is tested over Africa but is generic enough to be applied everywhere.






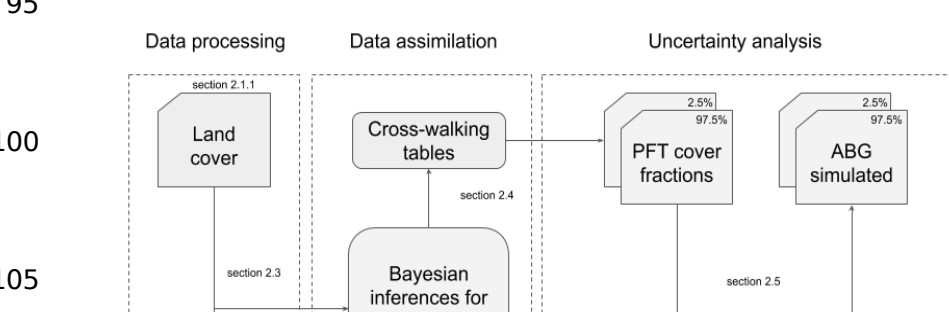

*Figure 1 Approach to assimilate the information held by AGB maps into PFT maps. Remote sensing AGB and land cover products are jointly assimilated to obtain cross-walking tables that can be used to make PFT maps. Owing to the uncertainty analysis in the data assimilation approach, an ensemble of cross-walking tables and PFT cover fraction maps can be produced. Subsequently, the land surface model ORCHIDEE can be run for different PFT maps to quantify the uncertainty from propagation the uncertainty from remote sensing products into a model simulation.*




## 2.2 Dataset products

### 2.2.1 Land cover map

ESA's Climate Change Initiative for Land Cover (CCI-LC) produced consistent global LC maps at 300 m spatial resolution on an annual basis from 1992 to 2015 [Defourny, P., 2019]. The typology of CCI-LC maps follows the Land Cover Classification System (LCCS) developed by the United Nations (UN) Food and Agriculture Organization (FAO), to enhance compatibility with similar products such as GLC2000, and GlobCover 2005 and 2009. The UN-LCCS typology was designed as a hierarchical classification, which allows adjusting the thematic detail of the legend. The "level 1" legend also

called "global" legend, counts 22 classes and is globally consistent and thus suitable for global applications such as creating PFT maps for LSM. The "level 2" or "regional" legend counts 43 classes which are not present all over the world and could be used in this study given its focus on a single continent, i.e., Africa (see section 2.2.3). In addition, the UN-LCCS partly overlaps with the PFTs used in climate models.

### 2.2.2 Aboveground biomass map

This study also makes use of a continental map of AGB of African savannas and woodlands for the year 2010 [Bouvet et al 2018]. The map has a 25 m resolution and is built from the 2010 L-band data of the Phased Array L-band Synthetic Aperture Radar (PALSAR) on the Advanced Land Observing Satellite (ALOS) satellite. Covering the African continent required about 180 data strips of which 91% were acquired between May and November 2010. The remaining 9% of the domain was filled with imagery from 2009 and 2008. The data have been processed by the Japan Aerospace Exploration Agency (JAXA)

using the large-scale mosaicking algorithm described in Shimada and Ohtaki (2010), including ortho-rectification, slope correction and radiometric calibration between neighboring strips, and by Bouvet et al. 2018 (multi-image filtering).





The continental AGB map was derived as follows: (1) stratification into wet/dry season areas in order to account for seasonal effects in the relationship between PALSAR backscatter and AGB, (2) the development of a statistical model relating the PALSAR backscatter to observed AGB, (3) Bayesian inversion of the direct model, to obtain AGB and its confident interval for pixels where no observations are available, and (4) masking out non-vegetated areas using the ESA-CCI Land Cover dataset (but see section 2.1.1). The resulting AGB map was visually compared with existing AGB maps [Saatchi et al., 2011, Baccini et al., 2012, Avitabile et al., 2016] and cross-validated with AGB estimates obtained from field measurements and LiDAR datasets [Naidoo et al., 2015]. Cross-validation revealed a good accuracy of the dataset, with an RMSD between 8 and 17 Mg·ha$^{-1}$. For more details on the creation and evaluation of the AGB maps see Bouvet et al. 2018.

### 2.2.3 Pre-processing

One known limitation of the original AGB map [Bouvet et al 2018] is the signal saturation and in some cases the decrease of the signal [Mermoz et al 2015] occurring in L-band SAR for AGB values higher than 85 t ha$^{-1}$. In order to overcome this issue, a second AGB map was created, based on two other ancillary datasets: a map of tree cover [Hansen et al 2013] and a map of tree height [Simard et al 2011]. Because of a coarser resolution from the tree height map (0,01°x0,01°, 100 ha) than the original ABG map (0,00025°x0.00025°, 0,0625 ha), the new biomass map has been rescale to 0,01° resolution. The rescaling will also drastically reduce the noise produced by PALSAR measurement artefacts (personal communication). The above-ground biomass was estimated by deriving an empirical relationship between biomass, available from airborne Lidar estimates, and the product of tree cover and tree height. The second version targets dense forest areas such as in the Congo basin (personal communication) and is used to adjust the AGB values at locations where signal saturation occurred (not published). The map used in this study is a composite of the two versions of the map by using the following rules:

- For broadleaved evergreen forests (UN-LCCS land cover type 50), flood forests (UN-LCCS 160), and closed broadleaved deciduous forests (UN-LCCS 61), the map based on tree cover and tree height was used because there is no AGB estimates in the map based on PALSAR.
- For broadleaved deciduous forests (UN-LCCS 60) the maximum between the two maps was used because its biomass ranged around the threshold of 85 t ha$^{-1}$ and may create truncated distribution.
- For the other land cover types, which typically have a biomass well below 85t ha$^{-1}$ the AGB value from the PALSAR map was used because it is considered more reliable than the statistical relationship between biomass, vegetation cover and vegetation height especially for the lower biomass.

Given the spatial domain of this study, only the 31 land cover types defined on the ESA CCI-LC map and present in Africa were retained. The complexity of the study was further reduced by removing all land types that cover less than 1% (304,158 km2) of the African surface or that contain less than 1% (i.e. 1,1 Gt) of the total AGB of Africa. Filtering retrained 15 out of





the 31 land cover types including bare land. These 15 land cover types (Table 1) represent 96% of the surface of Africa and 98% of its AGB.

One additional issue had to be dealt with: the spatial resolution of the land cover map (9 ha) largely differed from the resolution of the AGB map (0,01°x0,01°, 100 ha). As a consequence, each observational point on the AGB map is represented by 11,11 pixels on the land cover map. In order to simplify the overall data assimilation methodology (see section 3.2), we chose to use only AGB pixels (100 ha) which have a unique land cover type (i.e. pure pixels, in terms of LCC) . To this aim, the variety of land cover types across the 11,11 pixels within each AGB pixel (i.e., the number of LCC

present, *Vlct*) was calculated and only pixels where *Vlct=1* were retained. Although this criterion resulted in discarding 99% of the pixels, each of the 15 land cover types considered could be represented by at least 2000 pixels.

### 2.3 Data assimilation

#### 2.3.1 Linking land cover fractions and AGB

A linear model was used to relate the satellite-based AGB of a 100 ha pixel to the cover fraction of the satellite-based

vegetation types present at the same location. This relationship can be written as:

$$B_p = \sum_{i=1}^{nV} F_{p,i} \cdot Bref_i \tag{1}$$

where $B_p$, is the AGB at a given pixel $p$, $F_{p,i}$ is the cover fraction of the vegetation type $i$ (i.e. the generic plant functional type (PFT) use for land surface models, see section 2.1 - overview), $Bref_i$ is the reference AGB for the vegetation type $i$ and $nV$ is the number of vegetation types (i.e. number of PFTs) present in the pixel $p$. Given the number of unknowns ($nV$ being

usually above 1), equation 1 has many solutions; many of which have no biological meaning. The equifinality of this model can be reduced by arguing that the large difference in biomass between woody, herbaceous and non-vegetated ecosystems combined by their respective cover fraction explains the majority of the biomass at pixel level. Following this assumption, equation 1 can be simplified as:

$$B_p = F_{p,w} \cdot Bref_w + (1 - F_{p,w} - F_{p,b}) \cdot Bref_h \tag{2.1}$$

$$B_p = F_{p,w} \cdot Bref_w + F_{p,h} \cdot Bref_h \tag{2.2}$$

where $F_{p,w}$, $F_{p,h}$ and $F_{p,b}$ are the fractions cover for woody vegetation (i.e. woody PFTs), herbaceous vegetation (i.e. grassland and cropland) and non vegetated areas, respectively. $Bref_w$ and $Bref_h$ are the reference AGB of woody and herbaceous vegetation, respectively. Equation 2.1 is constrained by equation 2.2 (i.e. the total area coverage of each pixel), hence, $F_{p,h}$ in equation 2.1 can be substituted by equation 2.2 to obtain :

$$F_{p,w} + F_{p,h} + F_{p,b} = 1 \tag{3}$$



Although the model formalized in equation 3 no longer details which vegetation types i (i.e. PFTs) are present on each pixel p, it still has four unknowns and can, therefore, not be solved analytically. Nevertheless, a statistical solution is within reach if $F_{p,w}$, $F_{p,b}$, $Bref_w$ and $Bref_h$ are estimated from a population of AGB observations containing a number of independent

repetitions that largely exceeds the number of unknowns. In this study, over 2000 repetitions were available for each of the 15 land cover types that were retained following filtering (section 2.2.3). The statistical solution will thus consist of four mean parameter values (i.e., $F_{p,w}$, $F_{p,b}$, $Bref_w$ and $Bref_h$) for each of these 15 land cover types.

As described in section 2.2.3, the selection of homogeneous AGB pixels, i.e., which have a unique land cover class across the 11.11 (i.e. 121) underlying land cover sub-pixels allow us to rewrite the equation 3 as follow :

$$Bp_p = F_{lc,w} \cdot Bref_{lc,w} + (1 - F_{lc,w} - F_{lc,b}) \cdot Bref_{lc,h} \tag{4}$$

where $Bp_p$ now is the AGB of a specific land cover type lc and $F_{lc,w}$, $F_{lc,b}$, $Bref_{lc,w}$, $Bref_{lc,h}$ are the unknowns. The unknown parameters of the regression model (eq. 4) were estimated by using a Bayesian inference method. This approach has been chosen because it helps to synthesize various sources of information as well to propagate confident interval in the result of

our land surface model [Ellison 2004]. Bayesian inference requires, however, setting prior probability distributions for each of the unknowns, i.e., the biomasses and land cover fractions for each of the 16 land cover types. Given these prior probability distributions, Bayesian inference retrieves the posterior probability distribution for each of the unknown parameters.

### 2.3.2 Prior value distributions for $Bref_{lc,w}$, $Bref_{lc,h}$ and $Bp_p$

The pure AGB pixels were stratified according to their land cover type and for each land cover type the information contained in the distribution of the satellite-based AGB served to estimate the mean and standard deviation of the prior values of $Bref_{lc,w}$:

$$Bref_{lc,w} \sim N\left(\mu_{lc,w}, \sigma_{lc,w}\right) \tag{5}$$

where, $\mu_{lc,w}$ is calculated as follow:

$$\mu_{lc,w} = X^{th}per.\left(Bp_{lc}\right) \tag{6}$$

Where $Bp_{lc}$ is a vector containing $Bp_p$ values that belong to the land cover type lc and $X^{th}per$ denotes the 95th percentile for the woody cover types. This choice assumes that with the highest 95th percentile we select the AGB value of a pixel covered

only by woody vegetation (i.e. woody PFT) for the selected land cover type. In contrast to using in situ observations to





define $\mu_{lc,w}$, such approach offers the advantage to rely on a large ensemble of satellite-derived AGB observations and to be coherent with the following optimization.

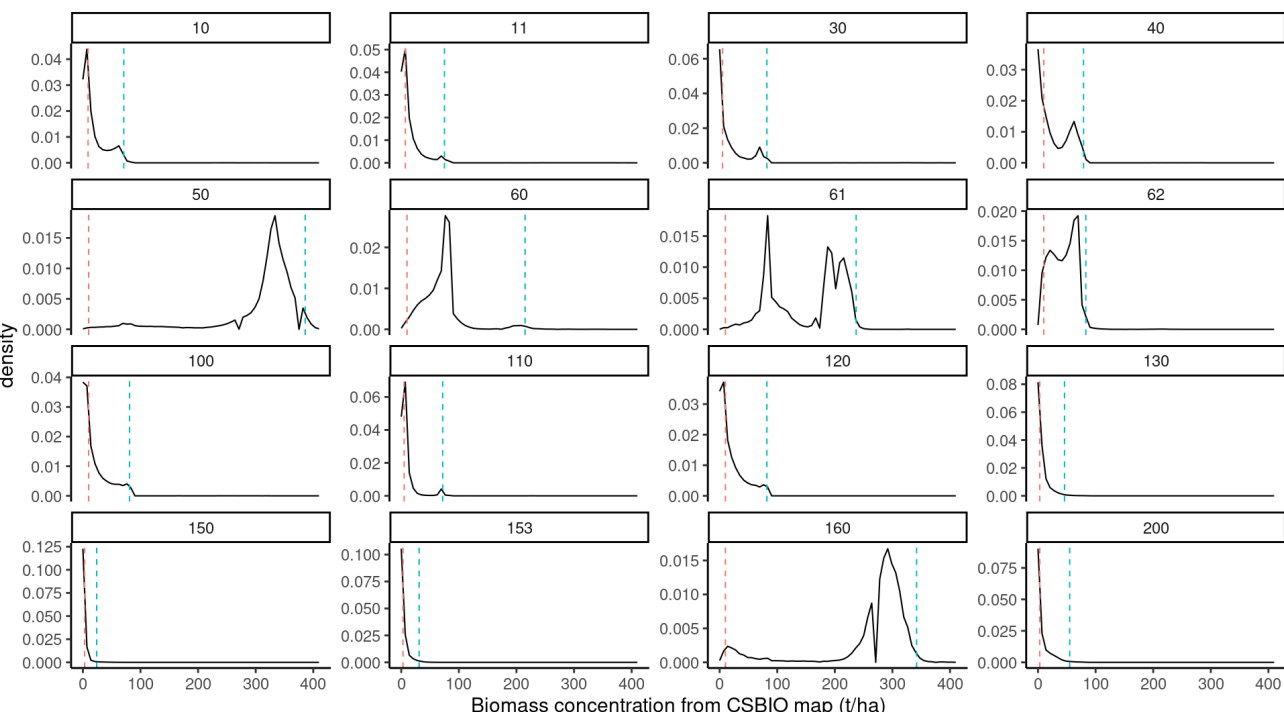

***Figure 2:*** *shows the distribution of the pure land cover pixel for biomass concentration $Bp_p$ for 15 selected land cover types + LCT 200.*
*The blue dashed line represents the $95^h$ percentile used as the prior estimate for the reference biomass concentration for trees $Bref_{lc,w}$.*
*The red dashed line represents the $50^{th}$ percentile also used as the prior estimate for the reference biomass*

Without any information about the variability of $Bref_{lc,w}$ , we choose to represent $\sigma_{lc,w}$ as:

$$\sigma_{lc,w} = \frac{\mu_{lc,w} \cdot 0.15}{4} \qquad (7)$$

Compared to $Bref_{lc,w}$, $Bref_{lc,h}$ is more difficult to assess from the satellite-derived data because it often shows bimodal distributions which may stem from biomass degradation or the presence of shrubs which biomass better resembles that of a grassland than a woody ecosystem (Fig. 2). We found that while the $2,5^{th}$ percentile is representing the lowest biomass for herbaceous ecosystem, the $50^{th}$ percentile seems to better describe $Bref_{lc}$, following the equation (6). Without any information about the variability of $Bref_{lc,h}$ , we choose to represent $\sigma lc,h$ as in equation (7).

Finally, $Bp_p$ comes with a measurement uncertainty that was thought to follow a normal distribution. Given that this measurement uncertainty is not known at the pixel level, an uninformative prior was set for the standard deviation $\sigma b_{lc}$ wich can varies between 0 and 200 t/ha :

$$Bp_p \sim \mathcal{N}\left(\mu, \sigma b_{lc}{}^2\right) \quad with \quad \begin{aligned} \mu &= Bp_p \\ \sigma b_{lc} &\sim \mathcal{U}(0, 200) \end{aligned} \qquad (8)$$





### 2.3.3 Prior value distributions for $F_{lc,w}$, $F_{lc,b}$

$F_{lc,w}$, $F_{lc,b}$ were defined as fractions of respectively woody vegetation and bare soil within a given land cover type, their values thus range between zero and one. For this reason, the probability distributions of the cover fractions were described by pseudo-normal distributions bounded by zero and one. The generalized Beta-distribution can represent a bounded pseudo-normal distribution and was used to describe the probability distribution of the woody and bare soil cover fraction:

$$F_{lc,i} \sim Beta\left(\alpha_{lc,i}, \beta_{lc,i}\right) \cdot \left(B_{up,i} - B_{lw,i}\right) + B_{lw,i} \tag{9}$$

Where i represent w (woody) or b (bare soil) described in the equation (5), $B_{up,i}$ and $B_{lw,i}$ represent the expected range (upper and lower values) of $F_{lc,i}$. A generalized beta distribution as two properties that can be exploited to estimate its parameters $\alpha_{lc,i}$ and $\beta_{lc,i}$ for each land cover type based on the mode ($\theta_{lc,i}$), the certainty of a beta distribution, i.e. $\omega_{lc,i}$ which is the inverse of the uncertainty, and the distribution boundaries $B_{up,i}$ and $B_{lw,i}$:

$$\alpha_{lc,i} = \left(\theta_{lc,i} - B_{up,i}\right)/\left(B_{lw,i} - B_{up,i}\right) \cdot \left(\omega_{lc,i} - 2\right) + 1) \tag{10}$$

$$\beta_{lc,i} = \omega_{lc,i} - \alpha_{lc,i} \tag{11}$$

where $\theta_{lc,i}$ is taken from a recent update of the CWT (see ORCHIDAS, with an expert-based update of the original CWT described in Poulter et al., 2015), $B_{up,i}$ and $B_{lw,i}$ are estimated based on the definition of the ESA-CCI-CL land cover classes and $\omega_{lc,i}$ was described by an uninformative uniform distribution and represents the trust we have in the current CWT:

$$\omega_{lc,i} \sim U\left(0, 200\right) \tag{12}$$

### 2.4 Confident interval propagation

### 2.4.1 Propagating the confident interval from the CWT into the PFT map

The posterior estimates of the cover fractions ($F_{lc,w}$, $F_{lc,b}$) will be directly used to make up a new cross-walking table. $F_{lc,w}$, and $F_{lc,b}$ values are then used to recalculate woody herbaceous fraction of each generic PFT of the CWT. In other words, we keep the original split of the different woody PFT defined in prior CWT and only rescale the total woody fraction to $F_{lc,w}$. Then we rescale the bare soil fraction based on $F_{lc,b}$ to finally rescale short vegetation PFTs (grass and crop) but using (1- $F_{lc,w}$ - $F_{lc,b}$).

Given that these posterior estimates come with a probability distribution, a probability distribution of the CWT could be made. In this study, the 2,5 and 97,5 percentiles of the posterior estimates were used to create two cross-walking tables that were then applied on the ESA-CCI-LC product to create two PFT maps that represent the 95% interval confidence of the ESA-CC-LC product, the AGB product, and the processing chain described in sections 2.2 and 2.3. The impact of the



confident interval on the PFT map was quantified for simulated above ground biomass and simulated albedo by running two simulations that only differed by the PFT map used to initialize the ORCHIDEE land surface model.

### 2.4.2 Description of the ORCHIDEE land surface model

ORCHIDEE (Krinner et al., 2005; Peylin et al. in prep.) is the land surface model of the IPSL (Institut Pierre Simon Laplace) Earth system model. As it is a land surface model, ORCHIDEEHence, by conception, it can be coupled to a global circulation model. In a coupled setup, the atmospheric conditions affect the land surface and the land surface, in turn, affects the atmospheric conditions. However, when a study focuses just on changes in the land surface ORCHIDEE rather than on the interaction with the atmosphere, it also can be run as a stand-alone land surface model. The stand-alone configuration

receives atmospheric conditions such as temperature, humidity, and wind, to mention a few, from the so-called meteorological forcing. The resolution of the meteorological forcing determines the spatial resolution which can cover range from , and can cover any area ranging from the global domain to a single grid point to the entire globe. ORCHIDEE uses nested time steps: half-hourly for, e.g., photosynthesis and energy budget, daily, e.g., net primary production, and annual, e.g., vegetation dynamics.

Although ORCHIDEE does not enforce a spatial or temporal resolution, the model does use a spatial grid and equidistant time steps. The spatial resolution is an implicit user setting that is determined by the resolution of the meteorological data. ORCHIDEE can run on any temporal resolution; however, this apparent flexibility is restricted as the processes are formalized at given time steps: half-hourly (i.e. photosynthesis and energy budget), daily (i.e. net primary production), and annual (i.e. vegetation dynamics). Hence, meaningful simulations have a temporal resolution of 1 min to 1 h for the energy

balance, water balance, and photosynthesis calculations. In the land-only configuration used in this study, the default time step for these processes is 30 minutes.

When an application requires the land surface to be characterised by its In this study the model was run with 15 PFTs, where the additional PFTs represented tropical and boreal C3 grasslands which both belong to the meta-class of C3 grasslands. When an application requires the actual vegetation, the vegetation will have to be prescribed by annual land cover maps.

These maps have to follow specific rules for the LSM to be able to read them. In the case of ORCHIDEE the share of each of the 15 possible plant functional types (PFTs)PFTs needs to range between 0 and 1 and be specified for each pixel. When satellite-based land cover maps are used as the basis for an ORCHIDEE-specific PFT map, the satellite-based land cover classification will need to be converted to match the ORCHIDEE specifications. As mentioned already above, this involves two steps: i) the derivation of generic PFTs from the satellite land cover classes (in our case the ESA-CCI-LC product)

through the CWT discussed in this paper and ii) the final mapping of the generic PFTs into the 15 ORCHIDEE-specific PFTs using additional information on the bioclimatic zones and the partition of grassland/crops into C3 versus C4 photosynthetic pathway  [Lurton et al., 2020].




**Table 1: Description of the 15 plant functional types (PFT) used in ORCHIDEE to represent global vegetation.**

| PFT | Climate | Vegetation type | Phenology class |
|-----|---------|-----------------|-----------------|
| 1 | global | NA | Bare soil |
| 2 | Tropical | Woody | Broadleaf evergreen |
| 3 | Tropical | Woody | Broadleaf deciduous |
| 4 | Temperate | Woody | Needleleaf Evergreen |
| 5 | Temperate | Woody | Broadleaf Evergreen |
| 6 | Temperate | Woody | Broadleaf Summergreen |
| 7 | Boreal | Woody | Needleleaf Evergreen |
| 8 | Boreal | Woody | Broadleaf Summergreen |
| 9 | Boreal | Woody | Needleleaf Deciduous |
| 10 | Temperate | Herbaceous | Natural (C3) |
| 11 | global | Herbaceous | Natural (C4) |
| 12 | global | Herbaceous | Managed (C3) |
| 13 | global | Herbaceous | Managed (C4) |
| 14 | Tropical | Herbaceous | Natural (C3) |
| 15 | Boreal | Herbaceous | Natural (C3) |

**2.4.3 Experimental setup**

ORCHIDEE tags 2.0 (rev 6592) was used to run tree simulations that only differed by the PFT map used to initialize the model. Each simulation consisted of a 110 years long simulation between 1901 to 2010 with climate reconstruction that matched the simulation years. $CO_2$ concentration was fixed to 299,16 ppm that corresponds to the 2010 concentration, the CRU-NCEP/v8 was used as the climate forcing.


Finally, differences in simulated above ground biomass and surface albedo between the 2,5% and 97,5% were compared to each other as well as to the satellite-based AGB map [Bouvet et al 2018] and satellite-based albedo (REF). Note that in this





study, the AGB by Bouvet et al (2018) was only used to constrain the cover fractions used by ORCHIDEE. Hence, the model-based estimate of biomass per unit area is independent from the AGB map as none of the information contained in that map was used in the ORCHIDEE model to simulate biomass per unit area.

### 2.4.4 Ecoregions

In this study, we choose to represent results related to the LSM simulation by subdividing the african continent into ecologically homogeneous regions. We choose to follow the rules defined in the works of Olson et al. 2001.

## 3 Results

### 3.1 Prior and posterior distributions estimates

#### 3.1.1  Prior distributions estimate for cover fractions and reference biomasses

Prior distributions for the cover fractions and reference biomasses were determined for all 15 land cover classes separately, nevertheless, three broadly different groups could be distinguished: (1) The 95[th] percentile of biomass distribution for each land cover belonging in the first group was so high, i.e., from 230 to 371 t ha[-1], that the land cover types in this group must come with a substantial tree cover., i.e., a woody cover fractions of 0.75 or more. Examples of this group are land cover types UN-LCCS 50, 61, and 160 (tree cover broadleaf types, Table 2). (2) Contrary to the first group, the 95[th] percentile of biomass distribution for each land cover type of the second group is so low, i.e., from <15 to 26 t ha[-1], that these land cover types must be dominated by grasses or bare soil, i.e., a woody cover fraction of 0.25 or less and a substantial bare soil cover fraction i.e. from 0.01 to 0.7. Examples of this group are UN-LCCS 130, 150 and 153 (grassland and sparse vegetation,Table 2). (3) The biomass of the third group falls in between these extremes representing mosaic land cover types like the UN-LCCS 10, 11, 30, 40, 60, 62, 100, 110 and 120. When taken over the African continent, the biomass distribution of these land cover types shows bimodal biomass distributions indicating considerable variability within these land cover types (fig. 2). The bimodal biomass distribution is backed by a large variation of woody cover fraction within a land cover type, for example, 0,15 to 1 for UN-LCCS 60 which represents woodland to dry savanna.

#### 3.1.2  Posterior distributions for reference biomasses, cover fractions, and certainty $\omega_{lc,i}$.

Most posterior distributions of the herbaceous and woody reference biomasses ($Bref_{lc,h}$, $Bref_{lc,w}$) are close to their prior distribution except for land cover types dominated by grassland as UN-LCCS 130 or by forests as UN-LCCS 50, 60 and 160. In these cases, the posterior woody cover fractions ($F_{lc,w}$) are approaching the boundaries of their distributions which is close to 1 for the forests dominated land cover types and close to 0 for the grassland dominated types.

For land cover types dominated by either woody or herbaceous species, the Bayesian optimization was more likely to adjust the prior reference biomasses rather than the prior cover fractions (given an overall higher sensitivity of the total pixel





biomass to $Bref_{lc,w}$ than to $F_{lc,b}$). For other land cover types, the Bayesian optimization played on both terms to adjust the prior cover fractions instead of the prior reference biomasses. Except for sparse vegetation land cover types, i.e., UN-LCCS 150 and 153, posterior bare soil cover fractions ($F_{lc,b}$) were similar to their prior distributions meaning that the proposed use of the biomass map did not result in knowledge gain concerning the bare soil fractions. For this reason, we did not use the

new bare soil fraction estimate when compiling the cross-walking table as explained in the section 2.4.1.

By comparing, the prior and posterior distribution of the woody cover fraction ($F_{lc,w}$) and its certainty ($\omega_{lc,w}$), two broadly different groups emerged (Table 2). In the first group, the posterior values of the first mode, $\theta_{lc,w}$, of the distribution of $F_{lc,w}$ largely agreed with the value picked from the original cross-walking table given the woody and herbaceous reference

biomasses. For these land cover types, $\theta_{lc,w}$ came with a widely distributed $\omega_{lc,w}$ i.e. close to a uniform distribution, suggesting that the confident interval of the prior did not drive its posterior estimate. In other words, the posterior $\theta_{lc,w}$ are not sensitive to a wide range of prior $\theta_{lc,w}$ values. This group contains UN-LCCS 60 and 62 (i.e., tree cover broadleaf). In the second group, posterior values of the first mode, $\theta_{lc,w}$, of the distribution of $F_{lc,w}$ largely disagree with the value picked from the original cross-walking table. The certainty shows a narrow distribution suggesting that the posterior $\theta_{lc,w}$ is strongly driven by

the confident interval of its prior value, reflecting that the most likely value for $F_{lc,w}$ is very different from its prior. This is the case for UN-LCCS 10, 11, 30, 40, 50, 61, 100, 110, 120, 130, 150, 153, 160.

Owing to the Bayesian approach, the woody and herbaceous fraction within each land cover type is no longer deterministic (as was the case with the previous generation of cross-walking table such as Poulter et al. 2015) but now comes with a distribution. This distribution is the outcome of propagating the confident interval on the retrieved parameters ($F_{lc,w}$, $F_{lc,b}$)

obtained from the Bayesian approach into the final product, i.e., the PFT cover fraction map. The 95% confidence interval was studied by comparing the 2,5 and 97,5 percentile of the distribution of woody fractions. Recall that for the construction of the cross walking tables the bare soil fraction was ignored for all PFTs except PFT 1 (see 3.1.2, Table 1). Hence, the herbaceous cover fractions mirror the woody cover fractions and are therefore not shown.

The mean change in forest cover fraction between the 2,5 and 97,5 percentile of the distribution of refined PFT maps over

Africa was 4,0% with a standard deviation between pixels of ±4,0%. At the ecoregion scale, the largest confident interval in forest cover fraction was found in the Congo basin with an average of 13,5% for the six concerned ecoregions e.i. ecoregions where LCT 50 is dominant, (Fig 3). Because our analysis did not change the ratio between grass and crop into the refined cross-walking table, there are no differences between crop and grass cover fractions.






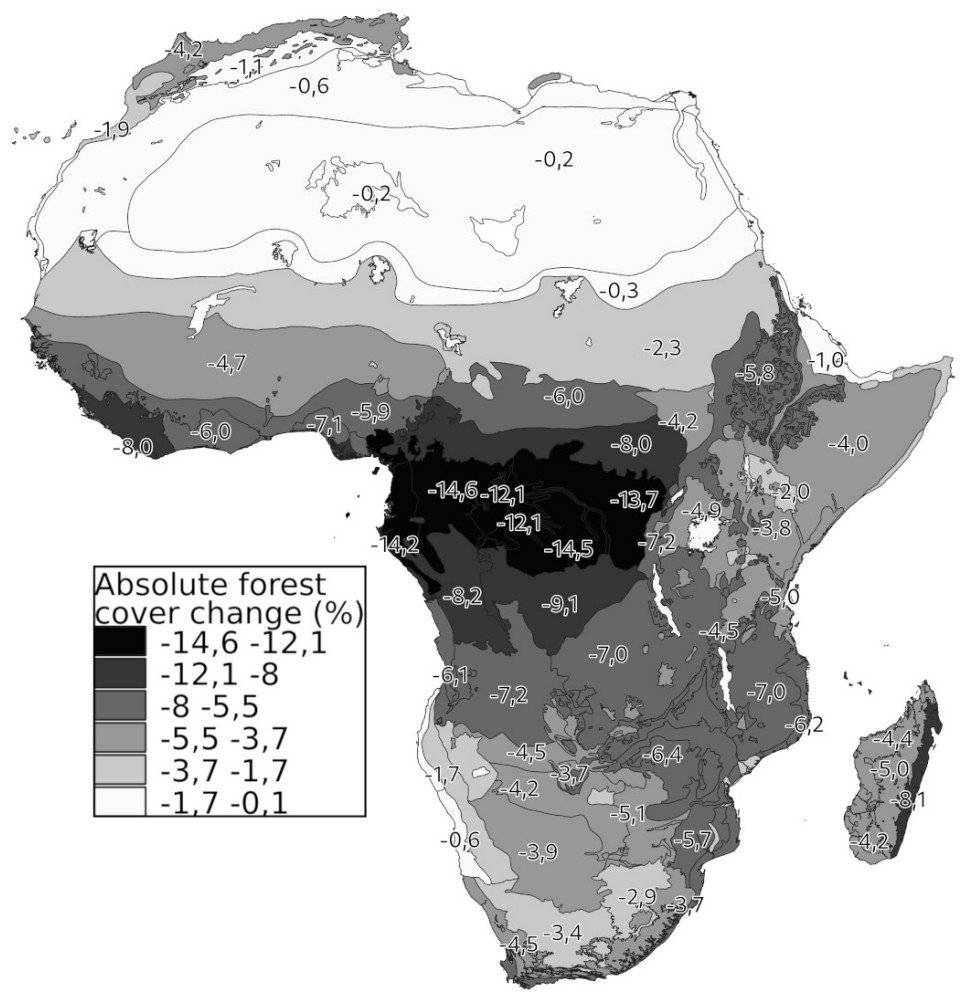

**Figure 3: A map of African's ecoregions showing the absolute change in forest cover fraction percentage (%) between the 2,5 and 97,5 percentile PFT maps. It can be seen as the uncertainty of the newly developed method. High values represent a large uncertainty in the estimation of the true forest cover fraction. The definition of the ecoregion has been taken from Olson et al. 2001.**







**Table 2: Short description, surface area (%), share in the continental biomass (%), prior parameters, and posterior median and confidence interval values for each of the 16 land cover types considered in this study. The numbering, description and surface area of each land cover type is based on the ESA-CCI product [Defourny, P., 2019], where its share in the continental biomass is based on a compilation of Bouvet et al 2018.** $\theta_{lc}$, $B_{up}$ $B_{lw}$, $\mu_{lc}$ **and** $\sigma_{lc}$ **represent the parameters describing the prior distributions of** $F_{lc}$ **and** $Bref_{lc}$. **Estimation of these parameters is detailed in section 2.3. Posterior distribution characteristics of land cover fractions** $F_{lc}$ **distributions and the certainty** $\omega_{lc}$. **For each land cover type and each parameter, the 2,5, 50 and the 97,5 percentiles are computed. The 50 percentile is a good approximation of the posterior** $\theta_{lc,w}$, **since the posterior distributions of** $F_{lc,w}$ **are symmetric. It is not true for bare soil cover fraction posterior distribution** $F_{lc,b}$ **and both** $\omega_{lc,w}$ **and** $\omega_{lc,b}$. **\* without any information in the short description, we choose to let the range be as wide as possible, \*\* the value has been changed from 0,4 to 0,6 to encompass the actual value used in the original CWT.**

| | Informations | | | Priors | | | | | | | | | | | | | Posteriors | | | | | | | | | |
|---|---|---|---|---|---|---|---|---|---|---|---|---|---|---|---|---|---|---|---|---|---|---|---|---|---|---|
| | | Surface | Biomass | $F_{lc,w}$ | | | $F_{lc,b}$ | | | $Bref_{lc,w}$ | | $Bref_{lc,h}$ | | $F_{lc,w}$ | | | $F_{lc,b}$ | | | $\omega_{lc,w}$ | | | $\omega_{lc,b}$ | | |
| id | UN-LCCS short description | area (%) | (%) | $\theta_{lc,w}$ | $B_{up,w}$ | $B_{lw,w}$ | $\theta_{lc,b}$ | $B_{up,b}$ | $B_{lw,b}$ | $\mu_{lc,w}$ | $\sigma_{lc,w}$ | $\mu_{lc,h}$ | $\sigma_{lc,h}$ | 2,5% | 50,0% | 97,5% | 2,5% | 50,0% | 97,5% | 2,5% | 50,0% | 97,5% | 2,5% | 50,0% | 97,5% |
| 10 | Cropland rainfed | 7,6 | 5 | 0,01 | 1* | 0 | 0,01 | 0,1 | 0 | 64 | 2,4 | 9 | 0,3 | 0,13 | 0,15 | 0,18 | 0,0049 | 0,0107 | 0,0271 | 2,6 | 12,8 | 42,6 | 6,5 | 101,7 | 195 |
| 11 | Cropland rainfed - Herbaceous cover | 3,2 | 3,3 | 0,01 | 1* | 0 | 0,01 | 0,1 | 0 | 55 | 2,1 | 6 | 0,2 | 0,15 | 0,18 | 0,21 | 0,0049 | 0,0107 | 0,0268 | 2,3 | 10,6 | 34,3 | 6,8 | 102 | 195,1 |
| 30 | Mosaic cropland (>50%) / natural vegetation (tree/shrub/herbaceous cover) (<50%) | 2,3 | 3,1 | 0,25 | 0,5 | 0 | 0,01 | 0,1 | 0 | 72 | 2,7 | 4 | 0,2 | 0,16 | 0,18 | 0,21 | 0,0048 | 0,0108 | 0,0302 | 3,2 | 30,4 | 91,6 | 5,4 | 99,3 | 195 |
| 40 | Mosaic natural vegetation (tree/shrub/herbaceous cover) (>50%) / cropland (<50%) | 2,2 | 1,9 | 0,5 | 1 | 0 | 0,01 | 0,1 | 0 | 72 | 2,7 | 9 | 0,3 | 0,27 | 0,3 | 0,34 | 0,0049 | 0,0108 | 0,0341 | 1,9 | 14,4 | 57,6 | 4,7 | 99,4 | 195,2 |
| 50 | Tree cover broadleaved evergreen closed to open (>15%) | 6,7 | 45,1 | 0,99 | 1 | 0,15 | 0,01 | 0,1 | 0 | 371 | 13,9 | 9 | 0,3 | 0,82 | 0,91 | 0,98 | 0,0049 | 0,0107 | 0,0265 | 3,1 | 20,5 | 88,8 | 6,8 | 100,9 | 195,1 |
| 60 | Tree cover broadleaved deciduous closed to open (>15%) | 4,2 | 8,7 | 0,7 | 1 | 0,15 | 0,01 | 0,1 | 0 | 112 | 4,2 | 9 | 0,3 | 0,63 | 0,68 | 0,73 | 0,005 | 0,0108 | 0,0294 | 8,1 | 60,1 | 98,2 | 5,7 | 100,5 | 195 |
| 61 | Tree cover broadleaved deciduous closed (>40%) | 0,4 | 1,8 | 0,85 | 1 | 0,4 | 0,01 | 0,1 | 0 | 230 | 8,6 | 9 | 0,3 | 0,6 | 0,65 | 0,71 | 0,0048 | 0,0108 | 0,0301 | 1,3 | 6 | 24,9 | 5,6 | 100,1 | 194,7 |
| 62 | Tree cover broadleaved deciduous open (15-40%) | 10,6 | 13,1 | 0,55 | 0,65** | 0,15 | 0,01 | 0,1 | 0 | 73 | 2,7 | 9 | 0,3 | 0,52 | 0,55 | 0,59 | 0,0049 | 0,0107 | 0,0283 | 8 | 59,3 | 98 | 6,1 | 99,9 | 194,9 |
| 100 | Mosaic tree and shrub (>50%) / herbaceous cover (<50%) | 1,8 | 1,5 | 0,6 | 1 | 0 | 0,01 | 0,1 | 0 | 71 | 2,7 | 9 | 0,3 | 0,14 | 0,16 | 0,18 | 0,0049 | 0,0108 | 0,0308 | 1,1 | 3,1 | 10,1 | 5,6 | 99,1 | 195 |
| 110 | Mosaic herbaceous cover (>50%) / tree and shrub (<50%) | 1,6 | 1,2 | 0,4 | 0,5 | 0 | 0,01 | 0,1 | 0 | 29 | 1,1 | 5 | 0,2 | 0,08 | 0,11 | 0,14 | 0,005 | 0,0108 | 0,0301 | 1,1 | 2,6 | 7,3 | 5,7 | 100,8 | 195,1 |
| 120 | Shrubland | 13,3 | 7,7 | 0,6 | 1* | 0 | 0,01 | 0,2 | 0 | 71 | 2,7 | 9 | 0,3 | 0,14 | 0,16 | 0,19 | 0,0041 | 0,0117 | 0,0504 | 1,1 | 3,1 | 10,4 | 5,8 | 99,7 | 195 |
| 130 | Grassland | 6,5 | 1,5 | 0,01 | 0,05 | 0 | 0,01 | 0,2 | 0 | 26 | 1 | 3 | 0,1 | 0,04 | 0,05 | 0,05 | 0,0028 | 0,0092 | 0,0194 | 1 | 1,7 | 4,6 | 12,7 | 110,2 | 195,6 |
| 150 | Sparse vegetation (tree/shrub/herbaceous cover) (<15%) | 1,6 | 0,2 | 0,1 | 0,15 | 0 | 0,7 | 1 | 0,5 | 15 | 0,6 | 3 | 0,1 | 0,03 | 0,04 | 0,05 | 0,5903 | 0,6627 | 0,6861 | 1,2 | 4,5 | 16,4 | 4,2 | 68,3 | 190,1 |
| 153 | Sparse herbaceous cover (<15%) | 1,1 | 0,1 | 0,01 | 0,15 | 0 | 0,7 | 1 | 0,5 | 15 | 0,6 | 3 | 0,1 | 0,06 | 0,07 | 0,07 | 0,5001 | 0,5069 | 0,5271 | 1,4 | 4,6 | 13,9 | 1 | 1,9 | 6 |
| 160 | Tree cover flooded fresh or brackish water | 0,7 | 3,5 | 0,75 | 1 | 0,15 | 0,01 | 0,1 | 0 | 327 | 12,3 | 9 | 0,3 | 0,79 | 0,85 | 0,91 | 0,0049 | 0,0107 | 0,0269 | 2,6 | 25,3 | 91,2 | 6,5 | 100,8 | 194,9 |

### 3.2.1 Comparison between the 2,5 and 97,5 percentile refined and the original PFT maps

The mean change in forest cover fraction between the refined PFT maps and the original PFT map are -11,7% (±14,8%) and -6% (±14,7%) respectively for the 2,5 and 97,5 percentile maps. Large disagreement between the refined and original maps was observed over the Somali Acacia-Commiphora Bushlands and Thickets and the Kalahari Xeric Savanna where forest cover fraction was found to be 32% lower in average for the refined PFT maps (Fig. 4a-b). The guinea forest showed a 6,5 to 12,5% higher forest cover fraction for the refined PFT maps. Several ecoregions e.g. easten guinean, Somali Acacia-Commiphora Bushlands and Thickets, show changes from the original PFT map of the same sign for the 2,5 and 97,5 percentiles which indicates that the Bayesian calibration strongly pushes a correction of the initial CWT into one direction.

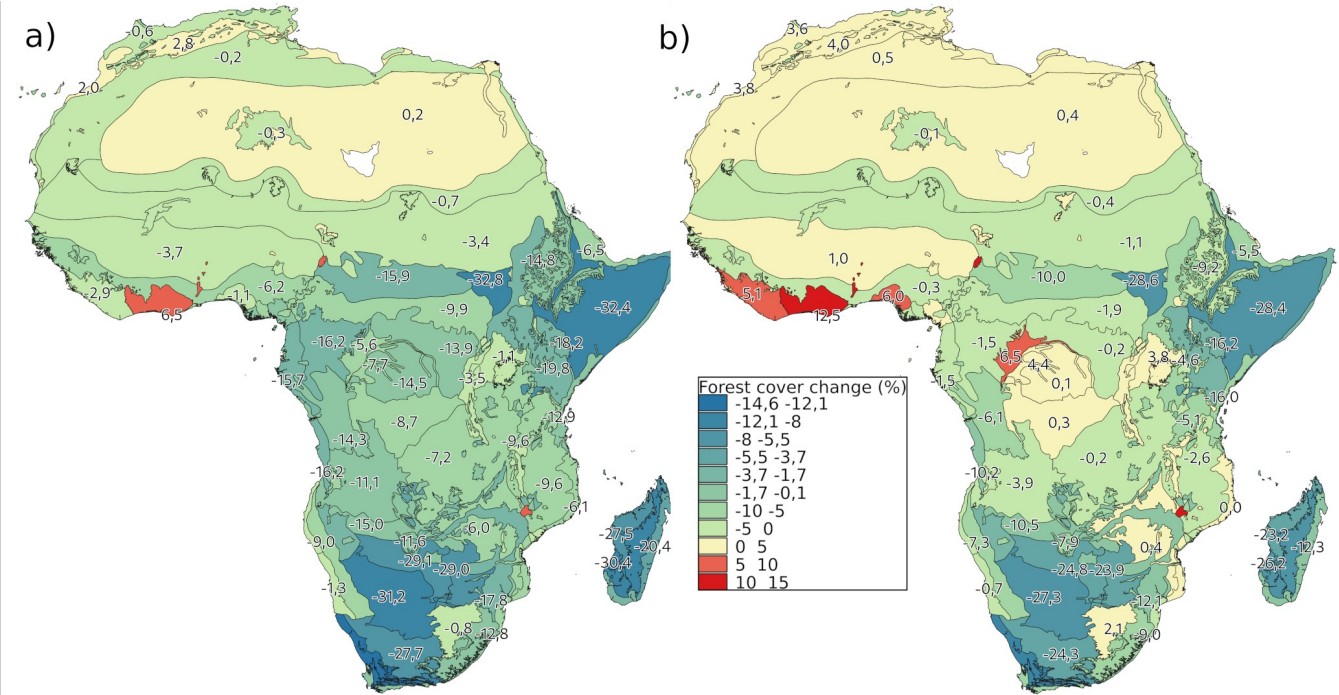

***Figure 4:*** *Forest cover fraction change in % relative to the original PFT map. Panel a) are for the 2,5 percentile PFT map and b) for 97,5 percentile PFT map generated by our new approach. PFT map generation is described in the section 2.4.1. The absolute difference between panel a) and b) are displayed in fig. 3. An ecoregion which shows the same sign of forest cover fraction change in both panel, strongly suggest a disagreement between the original and the new cover fractions i.e. woody, bare soil and herbaceous.*

### 3.3 Effect of the PFT maps on the biomass and albedo estimates

PFT maps are essential boundary conditions of land surface models because they condition the spatial distribution of various ecosystem states-properties (i.e. carbon content, albedo, water-carbon-energy fluxes, etc). When tested with the ORCHIDEE tags 2.0 (rev 6592)rev2.1 , the difference in biomass stock between the 2,5 and 97,5 percentile maps was 4,8 t.ha$^{-1}$ (not shown). The small differences in biomass between the original simulations (Fig. S2) can be explained by a modest cover fraction change from -12,2% to 1,3% in tropical rainforest (UN-LCCS 50 and 160), respectively for the 2,5% and the 97,5%





PFT maps compared to the original PFT map (Fig 4). When the PFT maps are propagated through ORCHIDEE, the aboveground biomass changes for these ecoregions result in -17 t.ha$^{-1}$ and 3,9 t.ha$^{-1}$, respectively from the 2,5% and the 97,5% PFT maps compared to the original PFT map (Fig. 5a-b). The difference in AGB estimates between the original and refined PFT maps was the largest over Madagascar (-52,3 t.ha$^{-1}$ in 2,5%), the Eastern Guinean Forests (+13,5 t.ha$^{-1}$ in 97,5%), and the Northwestern Congolian Lowland Forests (-33,9 t.ha$^{-1}$ in 2,5%)(Fig. 5a-b).

At the ecoregion scale, the largest difference between the albedo simulated with ORCHIDEE tags 2.0 (rev 6592)rev2.1 and initialized with the 2,5 and 97,5 percentile of the distribution of PFT maps and the simulated albedo for the original PFT map was found in Madagascar and ranged from 0,014 and 0,010 respectively from the 2,5% and the 97,5% (Fig. 5c-d). Nonetheless, the overall effect of the different PFT maps on the albedo is less than -0,0005.



**Figure 5:** *Confident interval propagation of the PFTs maps into AGB and albedo simulated by ORCHIDEE. The left panels represent the difference in AGB or albedo between the 2,5% CI PFT map and the original PFT map. The right panels represent the difference in the 97,5% CI PFT map and the original PFT map. a) and b) are above ground biomass change in t.ha$^{-1}$, and c) and d) are change in visible Albedo * 100.*





# 4 Discussion

## 4.1 Discretizing vegetation

Irrespective of the data products, the methods, and the model used, discretizing vegetation comes with its own challenges. Representing transitions of ecosystems by discretizing the vegetation into land cover type classes [Sankaran et al 2005] can lead to systematic errors since all pixels that belong to the same land cover class will get the same vegetation cover fractions (see 4.1.3) in the cross walking table. This approach articulates a key assumption underlying the PFT-approach, i.e., that only one life form survives and thus dominates the vegetation due to competition for nutrients, light and water [Hutchinson

et al. 1961]. However, the Savanna ecosystem, for example, is characterised by the coexistence of trees, shrubs and grasses which has been explained by interactions between vegetation, rainfall, fire, and browsing regimes [Eigentler and Sherratt. 2020]. This makes savannas one of the most difficult ecosystems to classify in a land cover type and subsequently convert it into a PFT map.

Over Africa, land cover classes such as shrubland (UN-LCCS 120) represent a wide range of ecosystems, from sparse xeric

shrubland composed of small bushes, e.g., Penzia incana (Thunb.) Kuntze, grasses, e.g., Sip Agrostis spp. such as found in Karoo desert, to dense thicket composed by succulent, e.g., Portulacaria afra Jacq. and spinescent shrubs (~3m tall) [Mills, et al. 2005]. Combining land cover types and biomass maps showed that the shrubland pixels in Africa often resemble sparse xeric shrubland than dense thickets. Improving the ability to simulate land surface properties of shrublands in a changing world, especially in Africa where shrub encroachment is an important land cover dynamic [Wigley et al. 2010, Buitenwerf et

al. 2012, O'Connor et al. 2014], is likely to benefit from a more detailed representation of shrublands in land surface models. A first step could be to represent shrubs as small trees, as was tested with the ORCHIDEE model for arctic ecosystems [Druel et al., 2017], but ultimately shrub density, largely controlled by precipitation [Rietkerk et al., 2002] should also be modeled.

Another major challenge with discretizing vegetation is how degraded ecosystems should be classified. From a modelling

point of view, they should be classified as the land cover type that occurred prior to the degradation and the cause of the degradation. e.g., fire, grazing, erosion, should be explicitly accounted for in the land surface. This ideal strongly differs from the current approach in which the degraded vegetation is classified as if the degraded vegetation is in its natural state. Even when having the correct PFTs, the current approach will fail to simulate the observed biomass if degradation occurred. As an alternative, the PFT map could duplicate all PFTs to distinguish between a PFT in its natural state and in its degraded

state. This approach in which degradation is accounted for in the PFT maps would, however, reduce degradation to a binary problem rather than addressing its continuous nature.

## 4.2 Knowledge gain from using the AGB map

In the absence of an AGB map, previous efforts to build cross-walking tables [Poulter et al. 2015] had to rely in part on expert knowledge. That generation of cross-walking tables can be considered as the best-available-knowledge in the absence



of AGB data or other information on the land surface cover. The method developed and demonstrated in this study mostly relies on data but comes with its own assumptions and statistical complexities. The key assumptions are that: (1) previous cross-walking tables [Poulter et al. 2015] are a reliable source to set the prior distribution for PFT cover, (2) the biomass map [Bouvet et al 2018] is a reliable source to set the prior distribution of the reference biomasses, and (3) the land cover classification contains homogeneous land cover types [Defourny, P. et al. 2019]. A key question is thus whether the added

complexity justifies the knowledge gained by jointly assimilating a land cover and a biomass map when producing a CWT? Ideally this question should be addressed by assessing the reduction of the confident interval associated to the posterior distribution of the PFT map when using the AGB map to constrain the CWT (in comparison to a prior when no AGB is used). However,  the present generation of CWT without AGB information, does not come with a distribution (except the attempt in Hartley et al., 2017), calling for an alternative approach to assess the knowledge gain. Given that the prior

distribution of the cover fraction was based on the previous CWT, the difference between the prior and the posterior distributions can be considered as the knowledge gained from using AGB information. Following this reason, the question we seek to answer is: "Is the cover fraction used by the original cross walking table falling outside the 95% confidence interval of our posterior estimate?"

If the answer is no, the biomass map is more likely in agreement with the previous effort to estimate the original cross

walking table. If the answer is yes, adding the information contained in the satellite based biomass maps is most likely in strong disagreement with the previous effort to estimate the original cross walking table. The original CWT has a global extent and the refined CWT is only valid for Africa. Therefore, knowledge gains should be carefully interpreted as they may reflect trade-offs that had to be made previously to construct a global rather than a regional CWT. Knowledge gains were assessed for: "croplands", "dense evergreen forests", "woodlands and savannas", and "xeric shrublands and grasslands"

separately.

### 4.2.1 Croplands (UN-LCCS 10, 11, 30, 40).

Despite the cover fraction of woody vegetation on croplands being close to none in the original CWT, this study found that the four land cover types associated with croplands, UN-LCCS 10, 11, 30, 40 are in fact covered by woody vegetation

ranging from 15% to 30% (range of median; Table 2). This large difference in the presence of woody vegetation on croplands is also reflected in the biomass data, which suggest two distinct but co-existing agricultural systems in Africa, i.e., one system with a low biomass and one around with a higher biomass.

The agricultural system with the low biomasses likely represents annually-replanted crops such as millet, sorghum, wheat,

sweet potatoes or cassava (FAO), with a maximum reported biomass between 10 and 15 t.ha$^{-1}$ for high-input cropping associated with commercial production of cassava and sweet potatoes. These values are in line with values estimated as reference biomass (see 2.3.2). Nonetheless, 97% of total cropland area Africa is rainfed  [Calzadilla et al. 2009] and most of





Africa's agricultural land is used for subsistence or small-scale farming associated with low-input cropping which explains why the actual average biomass estimate from the CESBIO map for cropland is between 1,98± 0,65 t.ha⁻¹ (Fig. 2) and thus

considerably lower than the potential production.

The high biomass agricultural system which is estimated at 58,6±10,2 t.ha⁻¹ in the CESBIO map (Fig. 2) likely includes plantations for coffee, rubber, fruits as well as shelter trees and forest remnants (FAO). Permanent croplands do not have their own land cover type in the UN-LCCS or in ORCHIDEE. In the absence of a dedicated PFT, these agricultural systems could be better simulated with a woodland fraction ranging from 13% to 34% (see table 2) , than is currently done with the

standard CWT. Although this could be an acceptable solution for biomass and albedo simulations, it will underestimate the agricultural production in the region.

### 4.2.2 Tropical rainforest (UN-LCCS 50, 160).

The woody cover fraction of tropical rainforest in the original CWT is close to 90%.  The original woody cover falls inside

the confidence interval of the posterior estimates, possibly because the 95% confidence interval is considerable and ranges from 80 to 100%. Strictly speaking, using a biomass map in addition to a land cover map did not result in knowledge gain concerning the cover fraction of tropical rainforest. Nevertheless, the considerable range in cover fraction indicates that many of the pixels classified as tropical rainforest do not all achieve the reference biomass of 371±14 t.ha⁻¹ (Fig. 2). The reference derived from the biomass map matches the AGB observed at field plots of intact forests in the Congo basin [Lewis

et al. 2013], the large variation in cover fraction for these land cover types may thus reflect wide-spread degradation of the forests in the region [Tyukavina et al. 2018] or an unrepresentative reference biomass [Kearsley et al. 2013].

### 4.2.3  Tropical moist deciduous forest/woodland/savanna (UN-LCCS 61, 60 and 62).

The woody cover fraction of the tropical moist deciduous forest ranged between 55% and 85% in the original CWT.

Refining the CWT by the use of AGB information narrowed this range to between 55% and 68%. For savanna (UN-LCCs 62) and woodland (UN-LCCS 60) the original cover fractions are within the refined 95% CI. For woody cover, the fraction of moist deciduous forest (UN-LCCS 61) decreased from 85% to 65%.

Although the reference biomasses used in this study are in line with previously reported values [Carreira et al. 2013], there are two ways to interpret the continuously decreasing biomass when moving from a forest, over a woodland towards

savanna. The original CWT considered these three land cover types as a single PFT. Differences between potential and actual biomass are the outcome of land use and are reflected in largely different woody cover fractions between these land cover types. The AGB map does not contain any evidence in support of this view and rather suggest that each of these land cover types comes with an own reference biomass, i.e., 197±30 t.ha-1 for dry forest, 62±14 t.ha⁻¹ for woodland and 22±12 t.ha⁻¹ for savanna (Fig. 2). Such a gradient in reference biomass could be justified by a climatological gradient. The large





range in reference biomasses compensated the range in actual biomass resulting in a relatively small range in woody cover fractions. Given the considerable degradation of these land cover types [Mitchard et al. 2013], the reality is likely a combination of degradation superimposed on a climate gradient.

### 4.2.4 Xeric shrubland (UN-LCCS 100, 110, 120).

The woody cover fraction of xeric shrublands and grasslands ranged between 40 and 60% in the original CWT. Accounting for the information contained in AGB map significantly decreased the woody cover fraction range toward 8 and 19%. Indeed, shrubs which represent a large part of the xeris shrublands were originally classified as woody vegetation for the ORCHIDEE model (i.e. when moving from the generi PFTs to the ORCHIDEE-specific PFTs; see section 2 and ORCHIDAS). This assumption is true from an ecological point of view but in a simplified world like in land surface models, xeric shrubland have an aboveground biomass that resembles that of cropland and grassland (Fig. 2). By overlaying the land cover type and aboveground biomass maps, 37% of the African shrublands were found to be degraded with a biomass of $2{,}7\pm1.5$ t.ha$^{-1}$, 54% were found to be intact with a biomass of $22\pm19$ t.ha$^{-1}$ and 9% of the shrublands are thickets with a biomass of $68\pm11$ t.ha$^{-1}$. This is in line with other aboveground biomass estimates from remote sensing products [Saatchi et al., 2011, Mitchard et al., 2013, Avitabile et al., 2016] and in situ measurements where shrublands, degraded thicket, and intact thicket in south Africa accumulated 3, 24 and 102 t.ha$^{-1}$ of biomass respectively [Mills, et al. 2005]. These findings suggest that in the model world, xeric shrubland is best represented by a large fraction of herbaceous plant functional groups, when the overall objective is to model AGB.

### 4.3  Consequences for land surface modelling

### 4.3.2 Which land cover types affect the biomass estimate?

In Tropical Rainforest (UN-LCCS 50, 160) and Deciduous Moist Forest (UN-LCCS 60, 61 and 62), the AGB obtained by driving ORCHIDEE with the original PFT map falls inside the confident interval simulated by driving ORCHIDEE with the refined PFT maps (Fig. 6). This is clearly not the case in the rest of the African ecoregions where the original PFT map systematically overestimates the 95% CI of simulated AGB obtained by using the refined PFTs maps. In general, the overestimation with the original PFT map is explained by an overall reduction in the forest cover fraction in the refined PFT maps.

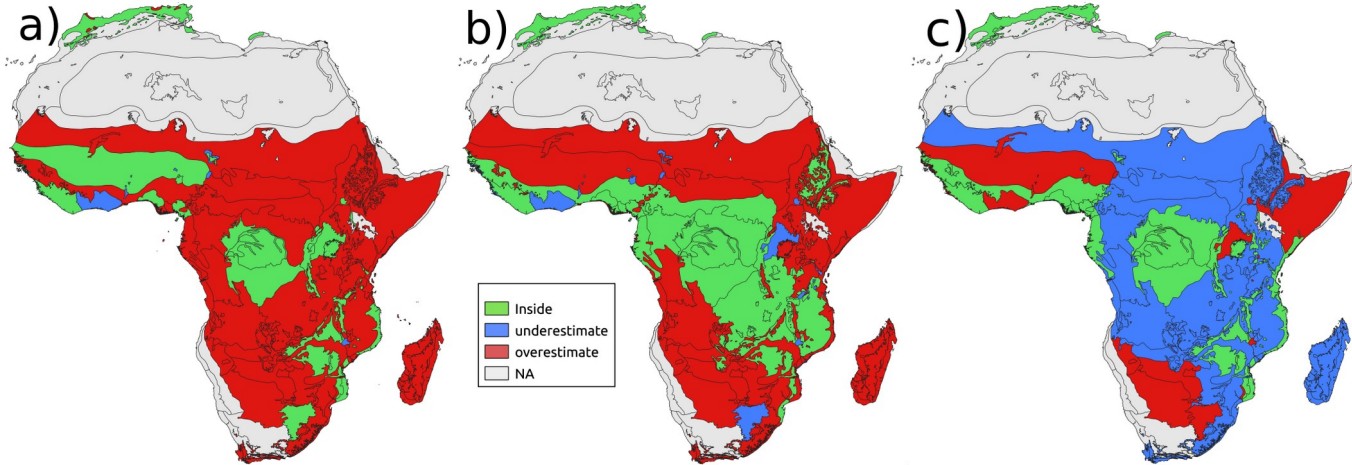

**Figure 6:** *Is the original PFT map fall inside the 95% confidence interval of the refined PFT maps (a) when used to drive ORCHIDEE to simulate AGB (b) and visible Albedo (c). For ecoregions colored in green, the AGB simulated making use of the original PFT map falls within the simulated 95% AGB range obtained by driving ORCHIDEE with the refined 2,5% and 97,5% PFT maps. Blue shows regions where the original PFT maps results in a significantly lower than the refined PFT maps. Red shows regions where the original PFT maps results in a significantly higher than the refined PFT maps. Grey indicates regions where the simulated AGB is less than 1 t.ha$^{-1}$.*

Compared to the original maps the refined maps prescribe a 28,4% to 32,4% lower tree cover for shrubland ecoregions like Somali Acacia-Commiphora Bushlands and Thickets and the Kalahari Xeric Savanna. Propagating these changes in cover fraction into simulated AGB, resulted in only small changes in carbon stocks (<1 t.ha$^{-1}$). This counter-intuitive result is explained by the growth processes simulated in ORCHIDEE. Under xeric climate conditions ORCHIDEE simulates low tree biomasses (< 2 t.ha$^{-1}$) because the low precipitation and subsequent plant water availability results in a continuous high tree mortality. In the refined maps, about ⅓ of the trees are replaced by grasslands which given the plant available water survive and even grow up to biomass of ~1,5 t.ha$^{-1}$.

Humid Mixed Cropland/Forest ecoregions like the eastern Guinean forests were systematically underestimated by the original PFT map. Indeed, compared to the original PFT map, the refined PFT maps prescribe a higher forest cover fraction for the cropland land cover type in order to include permanent tree crops like cacao, coffee and rubber plantation. Increasing the forest cover in humid regions results in a strong increase in the simulated AGB because in ORCHIDEE forests under these climates reach high (> 100 t.ha$^{-1}$) biomasses. To conclude, underestimating the forest cover in humid ecoregions will have a much larger consequence on the simulated AGB than overestimating the forest cover in xeric ecoregions.

### 4.3.3 Which land cover types affect the albedo estimate?

Overestimating the biomass is likely to come with overestimating the leaf area which in turn will result in underestimating the albedo because the reflectivity of leaves is often lower than the reflectivity of the soils it covers [Oke 2002]. Hence, Fig.





6c is expected to mirror Fig. 6b. This is not the case for the Somali Acacia-Commiphora Bushlands and Thickets and Sahelian ecoregions were replacing the forest by better growing grassland, both the biomass and albedo are decreasing.

## 4.4 Outlook

In this study a single biomass map was used as this enabled keeping the focus on the method itself. Nevertheless other biomass products are available [Saatchi et al., 2011, Baccini et al., 2012, Avitabile et al., 2016, Santoro et al., 2020] and could have been used. Repeating this study for each of these biomass products would add another source of uncertainty to the cross walking table. Owing to the method presented in this study, this uncertainty could then be propagated into the PFT map and all the way up to the simulated biomass, albedo -as done in this study for one biomass product- and other land surface properties. Considering different biomass products would give an insight of the impact of satellite-based biomass estimates on the discretisation of the vegetation and by extension surface properties as estimated by land surface models. Likewise, a single land cover map has been used in our analysis but other products are available as well [Copernicus, UN-spider, Li et al. 2020]. By using different land cover maps one could quantify the uncertainty in the land cover classification and propagate it to evaluate its impact on the simulated land surface properties.

Compared to other continents, the Africa vegetation has been documented by relatively few quantitative observations [Mills, et al. 2005, Saatchi et al., 2011, Asner et al., 2012, Réjou-Méchain et al., 2015]. Hence, it is the continent where remote sensing data could largely enhance our knowledge on the issue. Recent high-resolution satellite observations bear the promise to significantly reduce the confident interval around the aboveground carbon stock to estimate the $CO_2$ emissions from tropical forests [Hansen et al. 2013, Bouvet et al. 2018, Defourny et al. 2019, Buchhorn et al. 2020] but land surface models will need to be ready to routinely assimilate these data to fully benefit from the information contained in biomass maps. This study demonstrated one way of how satellite-based biomass data can help modelers to refine the initialization process by means of refining the cross-walking tables that are used to map land cover classes derived from satellite observations into PFT maps. Nevertheless biomass maps could be used for applications other than model initialization (this study), including model parameterisation and model evaluation.

The biomass map could be used to optimize model parameters related to growth, turnover and mortality to better simulate the vegetation biomass for the different PFTs. The evaluation stage could benefit from the biomass maps by benchmarking the model results against observed relationships between biomass-climate and biomass-land-use to better distinguish and simulate the difference between actual and potential biomass [Sankaran et al 2005]. Although the availability of several biomass products makes it possible to use one product to inform the cross walking tables and another product to evaluate the simulated surface properties, the magnitude of present-day differences between biomass products [Mitchard et al. 2013] is expected to result in major inconsistencies when different biomass products are used for different purposes (e.g., assimilation, parameterization, evaluation) into a single analysis. In this study, less than 0,01% (see 2.3.1) of the information contained in the biomass map was used to refine the cross-walking table and none was used to optimize model parameters. The simulated biomass (Fig. S2) remains, therefore, largely independent from the biomass map which implies that a single





biomass map can be used for land cover optimisation (as in this study), and in a second step for parameter optimization or model evaluation.

With an increase in resolution of the land cover map comes a decrease in the reliance on the cross walking tables. Cross walking tables will no longer be required once the resolution will be high enough (around 10 x 10 m) such that each pixel

contains a single vegetation type equivalent to a single PFT classification used by LSM [Li et al. 2020]. No longer having to rely on cross walking tables would likely reduce the confident interval of the PFT map. As there would no longer be a need to estimate woody and herbaceous fractions, there would no longer be a need for the information contained in the biomass map. It will then be feasible to solely use biomass maps to better parameterize the processes that contribute to simulating the reference biomass. It should be noted, however, that higher resolutions will not solve the basic challenge of discretizing

vegetation. High resolution land cover maps would split structurally complex ecosystems, for example savannas, into a pure forest fraction and a pure grassland fraction. This would overloop the interactions between the grasses and the trees which are among the defining ecological characteristics of a savanna.

Finally, we should note that other satellite-derived products than the AGB could be used to refine the mapping of the land cover classes into model PFTs (i.e., CWT). For instance, the global tree cover fraction map, at 30 meter resolution, from

620 Hansen et al. (2013) could also be used to refine the fraction of tree PFTs within each land cover class (as it was done in this paper with the AGB map).

## 5 Acknowledgements

This study was primarily financed by the French space agency, Centre National d'Etude Spatiale (CNES), under the specific program "BIOMASS-Valorisation", with the funding of Guillaume Marie and Cécile Dardel. The Marie Sklodowska Curie
Fellowship CLIMPRO (MSCA-Fellowship EU 895455) also contributed.

## 6 Data availability

▪   CESBIO African AGB map.Biomass map of Africa created by CESBIO can be downloaded on demand. It consists of a GIF file in which Africa is spatially discretized in pixels of 1x1km. The unit is a tonne of dry mass per hectare (t/ha). Contact person: thuy.letoan@cesbio.cnes.fr

        ▪   Land cover map is freely available here :  http://www.esa-landcover-cci.org.

        ▪   Ecoregion map use follows the work of Olson et al. 2001. This map is freely available here :
https://databasin.org/datasets/68635d7c77f1475f9b6c1d1dbe0a4c4c/





# 7 Code availability

- All R scripts and ORCHIDEE tags 2.0 (rev 6592) source code are available at :
  https://zenodo.org/badge/latestdoi/345907299 or DOI: 10.5281/zenodo.4785328

- ORCHIDEE tags 2.0 (rev 6592) code also available at :

https://forge.ipsl.jussieu.fr/orchidee/wiki/GroupActivities/CodeAvalaibilityPublication/
   ORCHIDEE_tags_2.0_gmd_2021_Africabrowser/tags/ORCHIDEE_2_1





## 8 Author contribution:

G. Marie, S. Luyssaert and P. Peylin designed the experiments and G. Marie carried them out. G. Marie developed the OPENBUGS model code and performed the simulations. G. Marie and S. Luyssaert prepared the manuscript with contributions from all co-authors.

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
