# Peer review of "Constraining a land cover map with satellite-based aboveground biomass estimates over Africa"

_Geoscientific Model Development, 2021_

## Author Response (AR1)

Referee 1 :

Dear Referee,
We would like to thank you for your constructive comments which will contribute to improving the manuscript.

Major comments:

- First, the authors need to test if the posterior distributions are very dependent on the prior distributions. If yes, this suggested that the optimization probably has not really worked

We can confirm that in our approach the prior distributions have a stronger influence on the posterior distributions. This should not necessarily be seen as evidence that the optimization did not work. Prior distributions should take all our knowledge into consideration and even if posterior distributions are dependent on the prior distribution it does not mean that the optimisation has not worked but just that the observations brought little constraining information to the inverse problem. We expect that changing the central value will result in different posterior distributions for some land cover classes whereas it will not affect the result for other land cover classes. Although the dependency is already discussed between Ln453 and Ln475 in the section 4.2 : "Knowledge gain from using the AGB map" in the revised manuscript, we will further elaborate on this issue by mentioning the referees viewpoint, and our interpretation of this dependency.

- The land cover types associated with trees and shrubs, i.e., 100, 110, 120 have very low posterior wood fractions (0.14, 0.08, and 0.14). This is not feasible, for the tree- and shrub- LC type, the wood fractions should be larger than herbaceous and bare soil fractions. Could you please provide the posterior woody and herbaceous biomass?

In the land cover type classification, land cover types 100, 110 and 120 represent very different ecosystems ranging from desert vegetation to closed thicket. As shown in figure 2, the biomass distribution of these three land cover classes is mainly determined by desert vegetation with a biomass of around 1 t/ha rather than a thicket with a biomass around 80 t/ha. In ORCHIDEE, we don't have a shrub PFT yet, so we have to choose whether we represent shrubs as a herbaceous PFT or as a closed forest PFT. For example, if we choose to represent shrub as a closed forest PFT, like in the original CWT, we risk to represent shrublands as a very small fraction of dense evergreen forest rather than a large fraction of sparse woodland (as it should be). The biomass of the forest will depend on the climate and the soil. This issue is discussed in *4.2.4: Xeric shrubland (UN-LCCS 100, 110, 120) of the new version of the manuscript (*between *Ln525 and Ln537)*.

- Second, I am wondering if the uncertainty of the AGB reference map would have large effects on posterior distribution? This is very important because different AGB products in some tropical regions have large differences.

We agree with this insight and we therefore suggested it as future work (section 4.4, Ln567 and 612). Given that the current manuscript already contains a lot of material, we did not include this analysis for the moment. In equation 8 the uniform distribution for $\sigma b_{lc}(0, 200)$ plays that role since for a given pure pixel biomass measurement ($Bp_p$), the standard deviation ranges from 0 to 200 t/ha. This accounts for the uncertainty of the biomass map we used. Future work should search for feasible approaches to account for uncertainties resulting from the fact that there are several observational-based biomass estimates available. This work would therefore have to deal with both uncertainties and biases.

Minor comments:

P2, Ln52: The first and second source of uncertainty looks the same, just interpreted in different ways.

We wrote "Current remote sensing technology does not enable distinguishing individual tree species, hence, vegetation is observed as land cover types [Defourny, P., 2019] which group vegetation with similar sensory characteristics. Remote sensing observations as well as classifying them in land cover types is a second source of uncertainties [Hansen et al. 2013, Mitchard et al. 2014, Hurtt et al. 2004]."

Rereading this sentence while keeping the referee's comment in mind, we still think the sentence is correct but might benefit from an extra line. The first source of uncertainty is that the signal we get from RS is already mixed because its resolution is too coarse to distinguish individual trees. The second source of uncertainty comes from the model architecture: the models are using PFTs which are a much coarser classification than species. We think our statement is correct because species-level RS data would still result in the PFT classification uncertainty. On the other hand, a model that would run at the species level would still be uncertain because of the mixed-species signal of the present day RS data. Given that the elimination of one source of uncertainty does not result in eliminating the other source, we conclude that these are two separate sources of uncertainty. We will clarify this issue in the introduction around L52.

P3, Ln65: Cannot find this reference in the reference list.
Thanks for noticing, we will add this reference in the reference list in the revised manuscript

P4, Fig 1: "ABG simulated" => "AGB simulated"
We will replace ABG by AGB in the new revision of the manuscript

P5, Ln 145: Please clarify the downscaling method.
The AGB map was downscaled by an average resampling method, i.e., computing the weighted average of all contributing pixels. To do so, we used the Gdalwarp function from GDAL (https://gdal.org/programs/gdalwarp.html). We will add this information in section 2.2.3 around Ln149 in the revised manuscript.

P7, Ln 170-171: Could you please show the locations of discarded and retained pixels? Are the retained pixels representative?

We can provide this information as a tiff file but in a figure formatted for an article, it would be unreadable, simply because the 1% remaining pixels are very small and will not be apparent from the figure. This is the reason why we choose to show the biomass distribution of the pixel instead (Fig. 2). We will add a sentence in section 2.2.3, around Ln173 clarifying to the reader that Fig. 2 shows the representativeness of the sampled pixels.

P10, Ln287: The first sentence is not completed.
We overlooked this issue while checking the manuscript before submission. We will complete the sentence in the revised manuscript.

P11, Ln306-307: The forcing data of the PFT map varied over time or not? If yes, the ESA CCI LC data starts with the year 1992. How to create the PFT maps before 1992?

Yes, the forcing data of the PFT map varied over time but our cross-walking table is only based on the 2015 ESA CCI LC data. In order to create PFT maps at other times than 2015, we use our cross-walking table especially crafted to this purpose. In the manuscript at line 116, the sentence may mislead readers about which data from ESA CCI LC is used. We will refine this sentence in the new revision of the manuscript.

P16, Ln396: What are the numbers in parentheses? Please clarify.
They represent the standard deviation of the mean change in forest cover fraction between the refined PFT maps and the original PFT map. A standard deviation has the same units as the mean for which it is calculated, hence, we added % after the standard deviation. Following the question of the referee we realized this might be misleading as it could be interpreted as the percentage of a percentage. We will remove the % symbol and report the standard deviation without units in the revised manuscript.

Section 3.2 and 3.3: I agree that comparing the percentiles of 2.5 and 97.5, which represent the lowest and the highest values, is very important. However, the mean and median values of distribution are more important than the upper and lower bounds. I think the authors should add the comparison of mean (or median) values. Another reason for this is that generally almost modelling studies only use the mean (or median) values to do evaluation, attribution, or projection assessments.

Our initial reasoning for only showing the 2.5 and 97.5% maps was that if these maps do not differ too much, it is not so informative to show the median map because it can't differ too much either. We agree with the referee that it is more straightforward to add a cross-walking table based on the median cover fractions and the subsequent simulation outputs in the section 3.2 and 3.3. We will do so in the revised manuscript.

P19, Ln 437-438: I don't really understand this. Using the PFT approach, we can set one type of forest PFT as 50%, and at the same time can set one type of grass PFT as 50%. Isn't it the coexistence of trees and grasses?

In the ORCHIDEE PFT's approach, splitting a pixel into 0.5/0.5 will create two independent ecosystems. By prescribing a forest PFT, ORCHIDEE will simulate a closed-canopy forest which is not at all representative of savanna trees. Also, because both PFTs are simulated independently the trees do not shade the grasses, do not reduce the throughfall for the

grasses, and trees and grasses won't compete for the same soil water nor for the same soil nutrients. Although the referee is right in saying that the PFTs coexist on the same pixel, the coexistence lacks the most basic interactions to qualify as coexistence or co-dominance as defined in the ecological literature (Sankaran et al. 2004). We will elaborate on "coexistence" in the revised manuscript around section 4.1 "Discretizing vegetation".

Referee 2 :

First, I think it is ethically questionable that a group of European scientists publish an analysis on Africa's natural resources without any involvement of African scientists. My motivation for this comment is the recent debate generated by the article of Misnany et al. (2020, https://doi.org/10.1016/j.geoderma.2020.114299), where they describe the concept of Helicopter Research as a form of neo-colonialism. Although the situation here is slightly different than that of collecting samples for scientific analysis, I still believe that similar questions can be asked in this manuscript. Why are African scientists not involved in this manuscript given the importance of defining PFTs and biomass for their own ecosystems? In their article, Misnany et al. highlight four negative aspects of this type of studies that do not involve local scientists: 1) Ignoring land ownership and disrespecting sovereignty. 2) Having little contribution to local science and development. 3) Promoting exclusivity—potential benefits to the studied country are often neglected, and further widens the gap between developed and developing countries. 4) Creating negative sentiments in local scientists towards international research.

We noticed the ethical objections of referee 2 and suggest that the editorial board of Geoscientific Model Developments seeks advice from scientists specialized in decolonization to settle this discussion rather than through a referee-author discussion. As the current journal policy does not stipulate the requirement of a geographical representation of the authors in line with the study domain, we will limit the discussion to the technical and scientific comments made by the referee.

Second, the use of the Bayesian approach is poorly developed. In particular, the choice of prior distributions is not consistent with formal theory for the specification of conjugate priors. For instance, the use of an uniform distribution U ~ (0, 200) for the prior distribution in equation 8 has no theoretical support; it leads to a distribution of biomass that extends to the negative side. In general, the formal Bayesian concepts for specifying hyperparameters are not used in this analysis. Therefore, I question the theoretical validity of the results presented in this study.

We agree with the referee that a wide uniform prior distribution in equation 8 will lead to unreasonable negative biomass values. To overcome this issue, we used a truncated normal distribution for biomass ($Bp_p$) as described by C(0,) (line 7) in the OPENbugs model (https://github.com/volarex84/R-script_African_biomass/blob/main/model_OPENBUGS.txt). Following the referee's comment, we realized that we forgot to update equation 8 in the text in line with the actual model used. We apologize for this oversight and will revise equation 8 to include the truncated normal distribution in the revised manuscript. You cna see the change in the new revision at line 218 and 258.

Minor comments

- Line 21 and thereafter. You use commas to separate decimal places. This is not standard notation in the English language.
We were advised by the editor to replace all decimal points by decimal commas as it represents the standard since 2018.

- Ln 145. ABG -> AGB.
Thank you for noticing, we will replace ABG by AGB in the revised manuscript.

- Ln 204. The correct spelling is 'confidence interval'. However, notice that in Bayesian statistics the correct term to use is 'credible interval' (see https://en.wikipedia.org/wiki/Credible_interval).
We will replace "confidence interval" by "credible interval" in the revised manuscript.

- Equation 5. Why do you assume a normal distribution? Biomass at the landscape level usually has a few sites with very large biomass. A distribution with a longer right tale would be more appropriate. Please provide a rationale for the selection of the gaussian distribution.

We agree with the referee that the biomass at the landscape level is not normally distributed but in the equation 5, we use the reference biomass which can be seen as the maximum potential biomass when soil, disturbance regime and climate are favorable, rather than the landscape biomass. The assumption underlying the normal distribution used in equation 5 is supported by the uncertainty around the reference biomass for a specific pixel which was derived from the CSBIO map where the uncertainty follows a normal distribution. The referee's comment made us realize that this assumption is not well explained in the manuscript. We added it in section 2.3.2 around Ln257-261 in the revised manuscript.

- Equation 8. This choice of prior for the standard deviation is unreasonable. It inevitably leads to negative biomass values.
see major comment 2

- Equation 9. I also see a problem with this choice of distribution. Once you pick one random value for one of the fractions, the other values are not independent. The Beta distribution alone cannot deal with this situation. The classical way to address this problem is with Dirichlet priors (see https://en.wikipedia.org/wiki/Dirichlet_distribution).

We agree with the referee, that using a multivariate beta distribution (also called Dirichlet distribution) is conceptually better than using two univariate dependent beta distributions as is now the case. We will revise our method and rerun all relevant analyses in order to include this suggestion in the revised manuscript.

- Ln 270. Why a reference to a study in preparation? There are hundreds of papers using Orchidee, and it has been described extensively everywhere else.
Although many authors cite Krinner et al 2005 when referring to the ORCHIDEE model, this reference is no longer accurate as almost all approaches described in this paper have been refined or replaced by a different approach in the 16 years that passed since its publication. The model version that is used in this manuscript is not accurately described by any of the

papers already published which justifies referring to a paper that is in preparation and that aims to describe this model version. The most accurate description of the ORCHIDEE model used in this paper can be found in Boucher et al 2020 (https://doi.org/10.1029/2019MS002010) but the description of the ORCHIDEE model is rather concise because that paper focuses on the entire IPSL Earth System model of which ORCHIDEE is the land surface component.

- Ln 287. Revise sentences.
We will revise the sentence in the new revision of the manuscript.

- Ln 306. Three?
We will replace "tree" by "three" in the revised manuscript.

- Ln 630. Without the African AGB map being publicly available this study would not meet reproducibility standards.
A hybrid map has been used for the analysis (https://github.com/volarex84/R-script_African_biomass/blob/main/Hybrid_biomass_map.tif), details of this map are given in section 2.x.x. The analysis presented in this study can be reproduced starting from this map.
The original biomass map is publicly available from:
https://www.theia-land.fr/en/product/african-biomass-map/

---

## Author Response (AR2)

Dear topical editor,
Thanks for reviewing our manuscript. Here the responses from your comment:

**1) First, I would like to refer to the ethical issue mentioned by Reviewer 2. I carefully reviewed the ethical guidelines of GMD and Copernicus, and the issue of 'helicopter research' is not addressed anywhere. I also consulted about the topic with the Chief Editor and we agree in that this is a topic that has not been discussed as an ethical issue for the journal or Copernicus. So, I cannot follow the advice of reviewer 2 to reject the manuscript based on this ethical consideration. We do not have an explicit policy for these cases.**
**Nevertheless, I think Reviewer 2 makes a valid point, and I would agree in that this manuscript could have benefitted from insights of local scientists that are familiar with the ecology of the African continent. However, for global or continental scale research, it is practically impossible to include scientists from all studied regions or countries.**
**My advice on this topic is simply to take these ethical considerations into account for future research, and see how local knowledge can be incorporated in large-scale analyses.**

Indeed, we should better take into account this aspect of our works. We will be more inclusive in our future researches.

**2) Reviewer 2 was correct regarding the use of the point symbol for separating decimal places. I guess I made the wrong suggestion in a previous round, and I apologize for it, but you were using inconsistent notation in the first place and continue doing so in this version (e.g. line 543). The correct separation symbol for the English language and in scientific publications is the point, e.g. 2.5 or 1.01. Please change this in all your figures, tables and text and be consistent.**

All the commas have been transformed in point in figures, tables and text.

**3) Line 193. Equation notation is inconsistent with referenced equations.**

Yes, you are right there was a mismatch between two revisions of mine. I rewrite equation 2 and 3 in order to retrieve the logical flow this section.

**4) Lines 296-297. 'Confident' and 'Credible' intervals are still used inconsistently. Please revise.**

Corrected

**5) Line 319. The term 'land surface' is repeated multiple times. Please review.**

I replace one land surface by "which" to increase the readiness of the sentence.

Best regards,
Guillaume MARIE